# Machine Learning to Identify Flexibility Signatures of Class A GPCR Inhibition

**DOI:** 10.3390/biom10030454

**Published:** 2020-03-14

**Authors:** Joseph Bemister-Buffington, Alex J. Wolf, Sebastian Raschka, Leslie A. Kuhn

**Affiliations:** 1Protein Structural Analysis and Design Lab, Department of Biochemistry and Molecular Biology, Michigan State University, 603 Wilson Road, East Lansing, MI 48824-1319, USA; jbemister34@gmail.com (J.B.-B.); wolf.alex9@gmail.com (A.J.W.); 2Department of Statistics, University of Wisconsin-Madison, Medical Science Center, 1300 University Avenue, Madison, WI 53706, USA; 3Department of Computer Science and Engineering, Michigan State University, 603 Wilson Road, East Lansing, MI 48824-1319, USA

**Keywords:** GPCR activity determinants, flexibility analysis, coupled residues, allostery, ProFlex, MLxtend, feature selection, pattern classification

## Abstract

We show that machine learning can pinpoint features distinguishing inactive from active states in proteins, in particular identifying key ligand binding site flexibility transitions in GPCRs that are triggered by biologically active ligands. Our analysis was performed on the helical segments and loops in 18 inactive and 9 active class A G protein-coupled receptors (GPCRs). These three-dimensional (3D) structures were determined in complex with ligands. However, considering the flexible versus rigid state identified by graph-theoretic ProFlex rigidity analysis for each helix and loop segment with the ligand removed, followed by feature selection and k-nearest neighbor classification, was sufficient to identify four segments surrounding the ligand binding site whose flexibility/rigidity accurately predicts whether a GPCR is in an active or inactive state. GPCRs bound to inhibitors were similar in their pattern of flexible versus rigid regions, whereas agonist-bound GPCRs were more flexible and diverse. This new ligand-proximal flexibility signature of GPCR activity was identified without knowledge of the ligand binding mode or previously defined switch regions, while being adjacent to the known transmission switch. Following this proof of concept, the ProFlex flexibility analysis coupled with pattern recognition and activity classification may be useful for predicting whether newly designed ligands behave as activators or inhibitors in protein families in general, based on the pattern of flexibility they induce in the protein.

## 1. Introduction

Recognizing the features of small, drug-like ligand molecules and protein structures that synergize to create an active protein state (binding to an agonist ligand) versus an inactive protein state (binding an inhibitory ligand) is essential to design drugs with predictable effects on the protein and organism. Much drug discovery research has focused on mimicking small molecule ligands of known activity (when available), either by incorporating very similar chemical groups that lead to cost-effective synthesis and favorable bioavailability and toxicity profiles, or by mimicking the three-dimensional volumes and chemical surface features of such molecules [1,2,3]. It is not uncommon for such molecules to bind the protein with moderate to high affinity, but not always with the activating or inhibitory effect that is sought. In this work, we focus on the other side of the interface, seeking a general method that can learn from a series of active and inactive structures in a protein family to identify the shared subset of protein features (without using ligand information) that are reliable indicators of whether the protein is in an active or inactive state. Identifying shared conformational changes, hydrogen bonds, hydrophobic contacts, and surface shape between protein structures has been carefully explored in GPCRs [4,5,6,7]. Sharing of features at an atomistic scale is dependent on conservation of the binding site and ligand type, however, and therefore fine-scale features are unlikely to be shared across complexes in a diverse family. Instead, we seek the signature of a shared flexibility mechanism, in the form of protein regions whose flexibility or rigidity in the ligand-bound state form recognizable patterns across active (or inactive) structures in the family. We then explore whether a small number of these intrinsic flexibility features can reliably predict whether a given protein is in an active or inactive state.

We present this methodology and apply it to individual structures of different class A GPCRs in a variety of conformations induced by small molecule agonists or antagonists, to discover hidden commonalities in flexibility/rigidity between the active (or inactive) states. The results provide new insights into how ligand binding to the orthosteric site (accessed from outside the cell) in this class of GPCRs can create flexibility changes adjacent to the transmission switch residues, which in turn undergo conformational changes acting as an on/off switch for binding intracellular protein partners and signaling to downstream partners. The shared changes in flexibility between GPCRs upon inhibitor or agonist binding also help distinguish key activity-relevant protein contacts of the inhibitors, and elucidate how inhibitors alter the network of intraprotein contacts to create biologically and pharmaceutically relevant responses.

For this analysis, we employed ProFlex, a successor to FIRST [8], an efficient and accurate tool for evaluating flexibility and rigidity within protein structures. Instead of analyzing conformational changes or dynamics, ProFlex analyzes the constraint network formed by covalent bonds, hydrogen bonds, and hydrophobic contacts to identify constrained (rigid) regions within a structure, as well as regions that are flexible and free to move due to the presence of fewer constraints. Coupling within rigid regions or flexible regions (e.g., cooperatively flexible loops) is also assessed automatically by ProFlex, with the rigid or flexible segments in a protein ranked from most rigid to most flexible. These segments may be as small as a few atoms (e.g., the cyclopropyl ring within proline) or as large as the entire protein, with no need for the user to partition atoms into artificial groups (e.g., main chain or side chains). ProFlex evaluates all covalent, hydrogen bond, and hydrophobic interactions and bond-rotational degrees of freedom within the system as a molecular graph on which bond and bond-angle constraints are counted, following the structural engineering theory developed by James Clark Maxwell, as extended to 2D and 3D atomic systems by Hendrickson, Jacobs, Thorpe, and Kuhn [8,9,10,11].

The goals of this study were twofold: (1) predicting with high accuracy whether GPCR structures are in active or inactive states; and (2) providing intuitive and human-interpretable insights into the underlying patterns associated with the predictions. To identify a subset of GPCR segments for making accurate activity predictions using a k-nearest neighbor classification model, we employed sequential and exhaustive feature selection algorithms. While exhaustive feature selection is guaranteed to find optimal feature subsets that maximize predictive performance, this combinatorial search problem is computationally intractable on large feature spaces. Hence, we employed sequential feature selection as a pre-filtering approach, which provides an excellent compromise in efficiency and effectiveness, to filter for feature subsets that maximize prediction accuracy of a k-nearest neighbor classifier before identifying the optimal feature subset via exhaustive search. All machine learning approaches employed in this study (exhaustive feature selection and k-nearest neighbor classification) are easy to use, yield intuitive results by highlighting the relative importance of predictive features, and are freely available from GitHub through the open source libraries MLxtend (http://rasbt.github.io/mlxtend/) and Scikit-learn (https://scikit-learn.org) [12,13].

The predictive motif ultimately identified by the ProFlex machine learning analysis in this work involves a tendency for the extracellular ends of helices 2, 3, and 5 and extracellular loop 1 surrounding the ligand binding site to be mutually rigid in inactive structures, as described in Section 3.2. The ionic lock, transmission, and tyrosine toggle conformational switch motifs identified by other researchers and reviewed in [14] involve different regions: the intracellular end of helix 3, a nonoverlapping segment of helix 5, and regions in helices 6 and 7. Thus, the ProFlex analysis provides new information and reveals commonalities in the ways different inhibitors induce an off state in class A GPCRs. This is important, because GPCRs comprise ~34% of all approved human drugs [15], and the goal of drug design for many GPCRs is to downregulate their activity. Beta blockers are one well-known class of inhibitory GPCR drugs, reducing blood pressure to substantially reduce cardiovascular risk and intraocular pressure in glaucoma to prevent retinal damage. Other GPCRs are targeted to control schizophrenia, allergies, and depression [16]. Our goal is to identify key regions in proteins that regulate their activity, on which researchers can then focus to improve drug design, as discussed in the *Conclusions*.

Two software utilities, BAT and BRAT (for B-value (Residue) Alignment Tool), have been developed and are available via GitHub (https://github.com/psa-lab/Protein-Alignment-Tool). BRAT facilitates identifying and visualizing the correspondence between user-defined sequence segments (such as ligand-binding residues) and residue numbers in one protein when aligned with a sequence-divergent homolog using Dali structural superposition [17]. BAT aligns and visualizes the temperature factor values (B-values), or other numeric properties recorded in the B-value column of Protein Data Bank (PDB) formatted protein structure files, across a number of user-selected, structurally aligned proteins.

## 2. Materials and Methods

### 2.1. Selecting GPCR Structures

Diverse class A GPCR structures in the Protein Data Bank (PDB; https://www.rcsb.org [18]) were selected for analysis, following these criteria: resolution of 2.9 Å or better, to ensure well-defined atomic positions and identification of appropriate non-covalent interactions; and no pairs of structures within the active or inactive sets with 80% or higher sequence identity, with the exception of PDB entries 2YDV and 3QAK, which are bound to significantly different ligands (Table 1). When possible, the same GPCR was represented by a structure bound to both an activating/agonist ligand and an inhibitory/antagonist ligand (as defined by the crystallographers). The resulting 18 inhibitor-bound GPCRs and 9 activator-bound GPCRs appear in Table 1, with the ligand, resolution, and R-factor (R value) data for each entry. Crystallographic R values measure the percentage difference in electron density when the data gathered from the diffraction experiment are overlaid with the electron density calculated from the atomic model that was fit into the electron density by the crystallographer, based on the known number of electrons associated with each atom type. A problem with this R(work) definition for assessing structural quality is that the refinement software used in structure determination is often designed to improve the fit between the model and the experimental electron density, which improves (lowers) the R(work) value but introduces bias. The R(free) value is used as a less biased measure of structural agreement between the fitted structural model and the electron density data. To calculate R(free), 10% of the experimental observations are removed from the dataset before refinement, and the refinement is then carried out with the remaining 90%. The R(free) value, also reflecting the percent difference in electron density between the experimental data and fitted model, is measured by comparing the electron density of the model fitted and refined to the 90% dataset with the experimental electron density calculated from the held-out 10% of the data. For an ideal structure, the R(free) value is close to the R(work) value, although typically it is higher. Lower values for both R(free) and R(work) are more favorable, showing greater agreement between the experimental data and the structure (https://pdb101.rcsb.org/learn/guide-to-understanding-pdb-data/r-value-and-r-free).

### 2.2. Defining Regions in GPCR Structures for Machine Learning

While ProFlex groups atoms that are flexible (or rigid) according to the natural partitioning of degrees of freedom in the protein chain following constraint-counting of covalent and non-covalent interactions in the bond network, machine learning with feature selection requires features that are consistently defined across the analyzed proteins. A natural feature representation, given the goal of identifying flexibility motifs in the protein associated with active or inactive states, is to segment the GPCR structures into small regions (Figure 1), and report the degree of flexibility in each region following ProFlex assessment. Accordingly, the extracellular (ECL) and intracellular (ICL) loops and canonical transmembrane helices (H1-H7) and C-terminal intracellular helix (H8) were numbered sequentially from the N-terminus to C-terminus, and then tabulated for each of the 27 protein structures. Each transmembrane helix was further segmented into three parts: the segment closest to the extracellular surface (e.g., H1.1 for helix 1), the most membrane-buried segment (H1.2), and the segment closest to the intracellular surface of the membrane (H1.3). This tripartite segmentation for transmembrane helices is based on prior observations that the extracellular, interior, and intracellular segments of transmembrane segments have different amino acid sequence attributes, and therefore it can be advantageous for structural predictions to consider the regions separately [19,20]. Figure 1 shows the resulting 29 segments considered in each GPCR structure (H1.1, H1.2, H1.3, ICL1, etc.) along with activity switch regions that have been characterized in class A GPCRs (the ionic lock, transmission switch, and tyrosine toggle; reviewed in [14]). The first extracellular loop in the GPCRs, preceding H1, was not included in the analysis. Its length and structure vary enormously across GPCRs, and this loop is often removed or altered in protein constructs prior to crystallization or fails to yield reliable atomic coordinates due to high mobility.

### 2.3. Performing and Interpreting ProFlex Analysis

To prepare PDB structures for ProFlex analysis, water molecule and hydrogen atom positions (which are absent or variably assigned between structures) and any ANISOU data records were removed. (These records encode anisotropic mobility data, with the same atomic coordinates repeated for the x, y, and z directions of motion; repeated atomic coordinates would be misinterpreted as new atoms by the software.) All ligands, as well as protein chains not relevant to the biological state of the protein (e.g., antibodies used to aid in crystallization), were removed before ProFlex analysis (v 5.2; https://github.com/psa-lab/proflex [8,11]). Hydrogen atom positions were then added consistently to all structures, in optimal orientations for hydrogen bonding, using the OptHyd method in the molecular mechanics package YASARA Structure (v 16.4.6; http://www.yasara.org [21]). Hydrogen atom positions may alternatively be assigned using other molecular mechanics software or Reduce (https://github.com/rlabduke/reduce). ProFlex was run as defined in the SiteInterlock protocol (https://github.com/psa-lab/siteinterlock with detailed documentation at https://psa-lab.github.io/siteinterlock/index.html [22]), without the ligand conformational search and docking steps preceding ProFlex, as the GPCR structures were analyzed without ligands. An appropriate hydrogen bond energy cut-off for ProFlex flexibility/rigidity analysis, defined by the HETHER routine (https://github.com/psa-lab/siteinterlock/blob/master/ scripts/proflex_hether.py) in the SiteInterlock protocol, was option C, the energy closest to (but less than) the level at which 70% of the protein residues were rigid.

Homology models for GPCRs contribute importantly to the field, given the difficulty of preparing native-like, pure membrane proteins for experimental structural determination. However, in past work, we noted that homology modeling does not always provide precise enough locations for the donor and acceptor atoms of hydrogen bonds, resulting in fewer identified bonds and underconstrained, overly flexible results from ProFlex. However, other aspects of a protein structure that are less dependent on positional resolution, such as the spatial location of different amino acid types and their clustering in protein structures, could also be good predictors of sites important for protein activation. Such alternative types of data as features can be used and tested as predictors with the same machine learning approach.

Aside from the structural resolution caveat, there is no fundamental limitation to the application of this ProFlex machine learning method to any protein family for which 3D structures and at least one known active case and one inactive case are available. That said, we would not advise mixing GPCRs from different families together, because the structures between GPCR families differ, as do their molecular partners and mechanisms of activation (especially for GPCRs that bind ligands in an extracellular domain). In different GPCR families, a different set of features may be key to activation. They can be unveiled by the machine learning feature selection approach described in Section 2.4., when trained on that particular family. Another aspect that can vary from family to family is whether the automatically chosen ProFlex energy level (HETHER option C, mentioned above) is appropriate for that particular family. This can be assessed most readily by a user who is knowledgeable about the protein family, by inputting to ProFlex a well-characterized active structure, then a well-characterized inactive structure, and visually identifying the energy level in the two ProFlex hydrogen bond dilution profiles (e.g., Figure 2A) that best identifies the known (literature-described) flexibility features that differ between the active and inactive states. Once that energy level is established, ideally by evaluating more than one protein in the family, it can be used as the ProFlex energy threshold for predicting the active/inactive state of other family members. Because known exemplars of active and inactive states are used by the k-nearest neighbor (KNN) classifier as the basis for predicting the activity of new structures, including more known examples may also improve the predictive accuracy.

The interplay between ProFlex and the KNN classifier used for prediction (Figure 2) begins with the hydrogen bond dilution (HBdilute) results from ProFlex. ProFlex includes all the hydrophobic and hydrogen bond interactions it detects in the protein structure using stringent geometric criteria [11]. The topmost data record (line) in the HBdilute results for PDB entry 2RH1, human β2-adrenergic G protein-coupled receptor (Figure 2A), shows the rigid regions (colored bars) and flexible regions (black lines) in the protein, from N-terminus to C-terminus, labeled by residue number along the top. The red bars indicate residues contributing to the largest rigid region in the protein, which at this energy level includes most of the structure except for a loop in the second half of the sequence, encompassing residues 231–265.

Using the HBdilute option, ProFlex then proceeds to analyze the protein at increasing hydrogen bond energy levels, mimicking the process of gradually heating the protein and observing how the energy-dependent hydrogen bonds break, one by one. Hydrophobic interactions, on the other hand, remain or tend to become stronger with moderate increases in energy [23]. Each time the breakage of a hydrogen bond dilutes the constraint network sufficiently that part of the protein becomes flexible (which ProFlex assesses quantitatively, using rigidity theory), a new line showing rigid and flexible regions is written in the HBdilute output (which is also provided in text format). Each new, separately rigid region appears as a bar in a different color (green, dark blue, light blue, and orange, in this case). The H-bond energy level for each line appears in kcal/mol in the second column from the left, with all hydrogen bonds in the current bond network being at least as strong as (equally or more negative than) this energy. The donor and acceptor atoms of the H-bond that was most recently broken are reported at the end each line. As a whole, the hydrogen bond dilution profile for a protein can be viewed as a profile of structural rigidity and flexibility from lowest energy (top line) to highest energy (bottom line), and used to identify the most persistently rigid or structurally stable regions in the protein, as well as how flexibility evolves in regions of the protein with increasing energy. Coupling information can also be derived from this output for the rigid region. For instance, helices H2, H3, H5, H6, and H7 all participate in one rigid cluster (red region) at the energy at which their helix labels appear in the center of Figure 2A, whereas H1 and H4 at that energy are separately rigid (green and light blue bars) and H8 has become almost entirely flexible (black line).

For the flexibility/rigidity analysis of all GPCRs, the highest (most negative) energy was chosen at which 70% or more of the residues were part of a rigid region. This 70% rigid level corresponds to a native-like state in which most GPCR helices and parts of the loops typically contribute to a large scaffold-like rigid region (l), with one or more of the helices and loops becoming separately rigid (s) or flexible (f). This energy level for β2-adrenergic receptor is indicated in Figure 2A by the helix labels H1, H2, H3, etc., appearing on the corresponding hydrogen bond dilution line. Figure 2B shows how the rigid regions at this energy level map onto the three-dimensional structure of β2-adrenergic receptor. The largest rigid region (red ribbon) is comprised by helix 2, helix 3, most of helix 5, and an extracellular short helix (top of figure, residues 177–187, not assigned a helix number since this helix is absent in other GPCRs) that is part of the mostly rigid loop connecting helices 4 and 5. Separately rigid regions appear in helix 1 (green ribbon) and helix 4 (light blue ribbon), and the position of the bound ligand (not included in ProFlex analysis) is indicated by the narrow tubes in green (carbon atoms), blue (nitrogen atoms), and red (oxygen atoms) behind the extracellular (upper part) of helices 3 and 4. Regions appearing in light grey in the structure are flexible at this energy level, corresponding to the horizontal black-lined regions in Figure 2A.

### 2.4. Machine Learning with ProFlex Features

To identify characteristic flexibility features and avoid overfitting when predicting protein activity, we focused on identifying the subset of features most likely to contain useful information (Figure 3). This was done in two ways. A profile of the frequency at which each segment (e.g., H1.1) was observed by ProFlex to be flexible, separately rigid, or part of the largest rigid region in active versus inactive structures (see Section 3) was used to identify features (e.g., ECL1l) with significant differences in prevalence (at least 25%) between active and inactive GPCRs. Those features showing the greatest difference in prevalence between active and inactive structures were considered sensitive features. As a second approach, feature selection algorithms were used to identify a subset of features showing the greatest discrimination between active and inactive proteins. Here, the term feature refers to the flexibility/rigidity state of each of the 29 segments in each GPCR structure. To identify useful feature subsets, we employed sequential feature selection (SFS) followed by exhaustive feature selection [24,25].

Exhaustive feature selection (EFS) evaluates all possible feature subsets that can be created from the original set (87 features). When evaluating all feature subsets, the goal is to select the one that maximizes a user-specified performance criterion, for example, the accuracy of a classification model trained to predict active/inactive protein structures. While this approach is guaranteed to find the optimal feature subset, it is computationally intractable due to the large number of feature subsets to be considered, unless the initial feature set is small. Even for small feature sets, the number of subsets can be prohibitively large. For example, the number of possible feature subsets of size 8 that can be created from a set of 29 features is more than 3 million (3,108,105).

Similar to EFS, sequential feature selection (SFS) reduces the original d-dimensional feature space to a k-dimensional feature subspace, where k < d. By contrast, SFS is a greedy search paradigm that constructs feature sets in an iterative fashion guaranteed to only improve the quality of prediction, but it does not evaluate every possible feature set. SFS is a computationally manageable alternative to EFS, and in our case was used as a feature-filtering step prior to EFS. This approach reduces the feature space to focus on features with the most predictive power. SFS exists in two modes, forward and backward SFS [25]. Backward mode SFS (Figure 4) removes features from the original feature set in an iterative fashion until the new, smaller feature subspace contains a user-specified number of features. In each iteration of the selection algorithm, an objective function is to be optimized. For instance, the objective function is commonly defined as minimizing the performance difference of a predictive model before and after removing a specific feature. In each round, backward-mode SFS eliminates the feature that causes the least performance loss upon removal [24].

Similar to backward mode SFS, forward mode SFS creates a feature subset of a user-specified size from the original set. Forward mode SFS starts with an empty feature set, adding one feature at a time (the feature resulting in highest predictive accuracy) until the feature set reaches a user-specified size, m, which is smaller than the number of features available for selection (e.g., the 87 flexibility values for structural segments in each GPCR). Since forward mode SFS starts with an empty feature set with features added one at a time, m iterations are necessary to obtain a feature subset of size m. In each iteration of forward mode SFS, the only features added to the training set are ones that were not added in prior iterations.

In addition, so-called floating versions of forward and backward mode SFS can explore a larger portion of the space of all possible feature subsets compared to SFS while still being computationally tractable [25]. In contrast to backward mode SFS, floating backward mode selection allows an already removed feature to be added back at a later iteration, if it improves the predictive performance of a classifier trained on this subset. Similarly, in floating forward selection, a feature that was previously added may be removed if this results in improved predictive accuracy.

To evaluate the performance of different feature subsets, a k-nearest neighbor (KNN) classifier, implemented using Scikit-learn’s KNeighborsClassifier model [13], was used in conjunction with leave-one-out cross-validation (LOOCV). In LOOCV, the classification model is applied n times to the each of the left-out test cases being predicted, and each training set consists of the remaining n-1 cases. In other words, of the 27 GPCRs, one is left out as the test case to be predicted by the KNN classifier, and the feature values and known active/inactive state of the other 26 GPCRs are used to train the KNN classifier, as shown in Figure 2. In each round, the model predicts whether the left-out case (represented by one GPCR feature set) corresponds to an active or inactive structure based on its nearest neighbors (feature sets plotted as points with activity labels) from the 27 GPCRs in the training set.

An example of using training set feature values as input to the KNN classifier appears in Figure 2C, where GPCR X is the new GPCR (or left-out training case) for which the active or inactive state is to be predicted. In the KNN classifier, the values of features for the training set cases are plotted on axes in a multi-dimensional space (up to eight dimensions, for up to eight features). In Figure 2D, a subset of three key features, H5.1l, H2.2s, and H3.1f, is being tested to predict activity. The corresponding feature values for each GPCR in the training set are plotted in this three-dimensional space. 2RH1, 3EML, and 2V2Y are plotted as values (1, 0, 0), corresponding to H5.1 being part of the largest rigid region, H2.2 not being a separate rigid region, and H3.1 not being flexible. These three proteins are all known members of the inactive class, in this two-class problem where a GPCR structure is defined as either active or inactive. Two known-active GPCRs, 3PQR and 2YDV, are plotted with their values (1, 0, 1). 5GLH, also active, is plotted with its (0, 1, 1) value, and the test case, GPCR X, is then plotted according to its feature values. The KNN classifier considers the k nearest training set neighbors of the test case, GPCR X, in this feature space, by computing the Jaccard similarity coefficient to measure nearness. The KNN classifier then predicts the class of GPCR X based on whether active or inactive training examples dominate as its nearest neighbors. Generally, an odd number of neighbors (odd k values) is considered to avoid the possibility of an equal number of neighbors from the two classes (to avoid tie-breaking schemes), and a series of different k values are tested. Class imbalance—the fact that more inactive GPCR structures than active GPCRs are available for training—must be addressed by the classifier in the choice of discriminatory features and an optimal k value; this is generally better than pruning examples from the training set, which loses useful information. The effect of class imbalance is considered again in the *Results*, in terms of the enhancement of predictive accuracy of the best feature sets relative to a dummy classifier, which simply predicts that all test cases match the dominant class in the training set (inactive).

After obtaining n predictions on the held-out data points in LOOCV for a given feature subset, the predictive accuracy for that set of features is computed as the percentage of predictions that were correct. Predictive accuracy was also measured by bootstrap cross-validation. For each bootstrap iteration for the 27 GPCR cases in the dataset, a random sample of 27 structures was selected from the GPCR dataset with replacement (meaning that a structure could be selected at random more than once). Every GPCR not in this training set was assigned to the *out-of-bag* test set. This bootstrap process, defining training and test sets for use with the selected feature set for KNN classification, was iterated 10,000 times, allowing the calculation of mean accuracy and standard error values. The most accurate feature sets and their leave-one-out and bootstrap accuracy statistics are summarized in Section 3.2.

Finally, the key features, meaning the superset of the SFS best-predictor feature sets from above, plus the features selected based on exhibiting at least 25% difference in prevalence between active and inactive GPCRs, were input to exhaustive feature selection. EFS enumerated all subsets of up to eight key features as input to the KNN classifier, to predict whether each GPCR was active or inactive (Figure 3, Step 4). Including more than eight features did not enhance prediction, consistent with the general statistical observation that overfitting is more likely to occur as the number of features approaches the number of cases being analyzed (27 in this study).

The general exhaustive and sequential feature selection methods outlined in this section can be combined with any machine learning algorithm for classification, and the specific MLxtend software implementation of SFS and EFS used in this study is compatible with any classifier implemented in Scikit-learn. We repeated the steps outlined in this section using generalized linear models such as logistic regression and a linear support vector machine (SVM) instead of KNN. Both logistic regression and linear SVM resulted in feature subsets with lower predictive performance compared with the KNN classifier, which is likely due to the linear models’ inability to capture the complex relationship between the input features and the class labels. A nonlinear radial basis function (RBF) kernel SVM was not considered in this study, as it requires extensive hyperparameter tuning and is thus prone to overfitting on a small dataset such as ours. Finally, we chose and focused on KNN as the primary classifier for this study, because it does not require extensive hyperparameter tuning and remains interpretable; for instance, predictions for new structures can be analyzed by querying and analyzing its nearest-neighbor structures in the existing dataset.

### 2.5. Comparing GPCR Regions and Numeric Properties with Alignment Visualization Tools

A challenge for GPCRs and many other protein families, given the evolutionary and functional diversity of sequences now available, is to identify which amino acid residues correspond between binding sites (or other regions of interest) when two sequences are homologous but cannot be aligned precisely (especially in less-conserved regions) by sequence similarity. This problem is easier to address for proteins with known three-dimensional structures, as considered here, because robust structural alignment tools such as Dali (http://ekhidna2.biocenter.helsinki.fi/dali/ [17]) are able to define which protein segments overlay significantly in 3D structure by comparing inter-alpha-carbon distance matrices rather than the amino acid sequences. The significance of the Dali structural alignment can be evaluated by its Z-score, measuring the number of standard deviations this alignment scores above a random structural alignment, taking into account the length and closeness of alpha-carbon overlay. Significant similarities have *Z*-scores above 2 and usually correspond to similar protein folds. From the resulting Dali structural alignment, the alignment of residues of interest to the user can be inferred. 

Two software utilities for highlighting sequence features of user interest, especially for proteins with regions of low sequence identity, have been developed in this work. These tools, BAT and BRAT (for B-value (Residue) Alignment Tool), are documented and available via GitHub (https://github.com/psa-lab/Protein-Alignment-Tool). As summarized in Figure 5, BRAT facilitates identifying and visualizing the correspondence between sequence segments of interest to the user (such as ligand-binding residues or extracellular loop regions) and residue numbers in one protein when aligned with a possibly sequence-divergent homolog, by using Dali structural superposition as input. BRAT alignment is written in HTML format suitable for publication or presentation, or comma-separated value (CSV) format suitable for further analysis, using single letter codes for the residues, with residue numbers labeled, and user-defined key residues highlighted. BRAT also supports automated definition of key residues based on the distance between residues and a user-specified ligand. The related BAT utility aligns and visualizes the temperature factor values (B-values) or other numeric properties recorded in the B-value column of PDB-formatted structure files, across two or more user-selected, Dali structurally aligned proteins. BAT writes the output of residues and correspondingly aligned B-values in CSV format, which can be read and analyzed further by spreadsheet tools such as Excel. These approaches provide more robust comparison between corresponding regions than a sequence-based approach for divergent sequences, such as the ligand binding sites or loop regions in GPCRs.

A meaningful comparison between key regions in two proteins (e.g., ligand binding or allosteric pathway residues) depends upon a reliable alignment of their protein sequences, rather than requiring 3D structures. For the present work, we focused on structure-based alignments because they allow definition of a clear correspondence between residues in protein regions where the sequence similarity is too low to allow confident sequence alignment. The helpfulness of structure-based alignment is particularly clear for the ligand binding sites of different class A GPCRs, which bind remarkably diverse ligands and therefore are not well conserved in sequence, while being substantially conserved in 3D structure. Structural alignment can define which residues between two proteins occur in the same position in the structure (or not). Sequence alignment methods that align one sequence to a multiple sequence alignment for the protein family, where the constituent sequences are chosen to reflect the protein’s evolutionary diversity, can partially address the challenge of aligning divergent sequences. This is because multiple sequence alignments containing many evolutionarily related sequences implicitly include information about the tolerance for different amino acid mutations and insertions or deletions at each position, which allows the alignment method to knowledgeably penalize for the presence of improbable residues or insertions or deletions at each position. For low-identity regions, it is still important to evaluate a *local* measure of the likelihood that each region of the sequence is correctly aligned before considering the residues in the proteins to be equivalent. Once such an alignment is available from any robust approach, formatting it as a standard Dali input (see documentation under https://github.com/psa-lab/Protein-Alignment-Tool) will allow BRAT and BAT to run successfully.

## 3. Results and Discussion

### 3.1. Identifying Key Flexibility Features for Predicting Activity

The frequency at which each structural segment occurs in a ProFlex-determined flexible, separately rigid, or largest rigid region in active versus inactive GPCR structures appears in Figure 6. Sensitive features to evaluate for predicting activity were derived from this profile, based on their large differences in frequency of occurrence between active (solid lines) and inactive structures (dashed lines). If two flexibility categories for a given segment (e.g., ECL2l and ECL2f) both showed large differences in frequency between active and inactive structures, the feature with the larger difference was selected as the sensitive feature. Features exhibiting a difference of 25-30% between active and inactive structures were ECL1l, H6.1s, and H8.2s. Features H1.3s, H3.1f, ECL2l, H5.1l, and H7.2l all differed between active and inactive cases by 30-40%, while H7.1l differed by 44%. H2.1s and H2.2s were the most discriminatory features, exhibiting 50-55% difference between active and inactive structures. Additional features selected by the forward or backward sequential feature selectors as most discriminatory between active and non-active cases were: H1.2f, ICL1f, H2.1f, H3.3f, ICL2f, H4.1l, H5.2f, H6.1f, and H7.3s; and H2.3s, H2.1l, ECL1f, H1.3l, and H2.2s.

### 3.2. Accurate Classification of GPCR Activity Based on the Flexibility of Key Regions

The top-performing four feature sets for predicting the activity of GPCRs in KNN classifier cross-validation appear in Table 2. Interestingly, a subset of four flexibility features, H2.2s, ECL1l, H3.1f, and H5.1l, were common to the top four feature-based predictors (96.3% leave-one-out and 79.6% bootstrap accuracy; top line in Table 2). Predictive accuracy was enhanced slightly by adding two features, H2.1s and H6.1f (second line in Table 2), to the above four. Figure 7 visualizes the spatial relationships between the top four structural flexibility features in a class A GPCR, β2-adrenergic receptor, where they were found to surround the ligand binding site.

### 3.3. Patterns of Flexibility and Correlation between Activity-Predicting Features

The features ECL1l and H5.1l were most important for predicting inactive states, followed by H6.1f, based on their enhanced occurrence in inactive GPCRs. On the other hand, active GPCRs were associated with greater flexibility in the key regions, specifically the presence of H3.1f, H2.1s, and H2.2s; these three features were never observed in the inactive GPCRs. While up to eight features were included in the feature sets sampled exhaustively as input to the classifier, none of the top predictors included more than six features. The best-performing feature-based classifiers (Table 2) were well-balanced between features associated with active states versus inactive states. A dummy classifier that always predicted structures as inactive (comprising the dominant class, 18 of the 27 GPCR structures) was used to assess the gain in accuracy from using sensitive feature selection plus SFS with the KNN classifier. Table 2 indicates that the best feature-selection predictors yielded 30% higher leave-one-out predictive accuracy and 21% higher bootstrap accuracy than the dummy classifier (bottom line), while also having 3% less variability in bootstrap accuracy (as measured by standard error).

How can these concepts and methods be applied to an individual GPCR, to help define residues that contribute to activation or inactivation? Here, we focus on one of the best-studied GPCRs, rhodopsin, given structures of its inactive (PDB entry 1GZM) and active (PDB entry 3PQR) states. Of the six key flexibility features identified here across class A GPCRs, two differ significantly between inactive and active rhodopsin states. ProFlex results show that the H2.1 region (the cytoplasmic third of helix 2) is part of the largest rigid region in inactive rhodopsin (known as opsin), while its initial residues are flexible in the active form. Secondly, the H6.1 region (the cytoplasmic third of helix 6) is separately rigid in opsin, while being flexible in active rhodopsin. These flexibility changes are consistent with the trend of key regions in class A GPCRs, as a whole, to contribute to the largest rigid region (the protein scaffold) in the inactive state while exhibiting increased flexibility and uncoupling to other regions of the protein in the active state (Figure 7). How do the ProFlex results compare with experiments characterizing the rhodopsin transition between inactive and active forms? H6.1 includes the ionic lock residue E247, which forms a salt bridge with R135 in opsin but not in active rhodopsin. The loss of the ionic lock interaction is consistent with the ProFlex observation of increased flexibility in H6.1 upon activation. Secondly, the cytoplasmic end of helix 6 (H6.1) is observed to hinge towards helix 5 when comparing the active and inactive structures [26], which is consistent with the increased flexibility of H6.1 found by ProFlex in the active state. Narrowing down the most important protein flexibility transition sites for activation from the plethora of conformational changes observed between crystal structures is a valuable application for ProFlex machine learning. This can suggest a much more focused set of experiments—to test H6.1 hinge residues, for instance—as well as indicating which flexibility transitions are shared with other class A GPCRs.

We also asked: To what extent were the most predictive flexibility features correlated? For instance, when ECL1 was observed to be part of the largest rigid region (resulting in feature ECL1l), was adjacent H3.1 flexible (H3.1f) or not? Correlation analysis can help us understand whether the flexibility features work together or are relatively independent in influencing GPCR activity. To address this, Figure 8 shows pairwise correlation of the six features in the most accurate predictor (Table 2). The only highly correlated features were H2.1s and H2.2s; in structures where the N-terminal segment of helix 2 is separately rigid (in the H2.1s state), the central segment of helix also tends to be separately rigid (H2.2s), with a correlation coefficient of 0.78. Seventy percent of the H2.1s and H2.2s occurrences are in active GPCRs. All other feature pairs in Figure 8 have absolute correlation values less than 0.45. Thus, most predictive features behave fairly independently of each other, while together being good predictors of an active or inactive GPCR state. 

### 3.4. Comparison with a Crystallographic Measure of Flexibility for Active and Inactive GPCRs

We then evaluated whether crystallographic thermal mobility (B-value) data present in PDB files could provide an alternative way of identifying regions that differ significantly in flexibility between active and inactive GPCRs. Figure 9 shows the B-value traces for a representative sample of three active and three inactive GPCRs; including traces for more structures made it difficult to visualize trends. In the inactive structures, two of the three (2VT4 and 3ODU) have similar, almost overlapping B-value traces, whereas 2RH1 shows relatively low variation in B-values from N-terminus to C-terminus. The pattern observed from the two similar inactive traces is that the loops ICL1, ICL2, ECL2, ICL3, and the N- and C-termini of the proteins are more flexible than the helical regions, while ECL1 is more rigid (consistent with ProFlex analysis). This is true of many protein structures, both membrane and soluble.

When the active structures were analyzed, the three B-value traces (5C1M, 5GLH, and 3QAK) had very different B-value scales. If this difference in B-value baseline was ignored and the regions of variation within each protein were considered and compared, ICL1, ICL2, ECL3, the N- and C-termini, and regions roughly corresponding to ECL1 and ICL3 were found to be more flexible than the rest of the structure in at least 2 of the 3 active proteins. ECL2 was missing (had undefined coordinates) for all three structures, likely indicative of very high mobility, and thus is a potentially useful signature of the active state. However, the loop and N- and C-terminal high-mobility features were all in common between active and inactive structures, aside from the relatively low B-values observed for ECL1 in inactive states compared with a high B-value, flexible state within or preceding this loop in active structures. Overall, it would be more difficult to predict activity-associated regions from B-values because of their inconsistency in baseline magnitudes, and the limited variation observed within some of the proteins. This is likely due to the crystals being held at different temperatures during data collection, and the GPCRs packing differently in their crystal lattices (some more constrained than others). Additionally, different methods were used for refining the structures, which can strongly affect B-values. Another consideration is that B-values measure mobility of atoms around their average coordinates. Thus, an internally rigid helix with a hinge at one end, which is able to freely swing like a lever arm, can show large B-values at the swinging end of the helix and much smaller B-values near the hinge. In contrast, ProFlex measures flexibility based on internal rotational degrees of freedom, rather than the Cartesian representation of mobility used for B-values. ProFlex would therefore label the hinged, swinging helix to be separately rigid rather than flexible. Another difference between B-values and ProFlex evaluation of flexibility is that information on coordinated motion within flexible or rigid regions is an automatic feature of ProFlex analysis, whereas inferring coupling information from B-value data is computationally and memory-intensive, requiring principal component or essential dynamics analysis.

### 3.5. Using the BRAT and BAT Alignment Visualization Tools to Identify Corresponding Sites and Quantitative Features from Structural Alignments of Sequence-Diverse Homologs

A challenge for working with a diverse set of homologs, such as the hundreds of GPCRs present in humans, is to define corresponding regions such as ligand binding sites or flexibility motifs when the sequences are too divergent to align confidently. For this purpose, we developed two Python alignment visualization utilities (https://github.com/psa-lab/Protein-Alignment-Tool). The first is BRAT, which starts with a Dali pairwise structural alignment and then highlights regions of user interest, such as ligand binding residues, which can be defined by the user as residue ranges in a reference structure (e.g., 2RH1), or as the set of residues within a user-defined distance X (in Å) of a ligand atom (as computed by BRAT). The output is a BRAT-formatted pairwise sequence alignment in either comma-separated value (CSV) or hypertext markup language (HTML) web-viewable format, which the user can further edit/annotate and incorporate in publications and presentations. An example of BRAT output (Figure 10) shows the signature of six flexibility features that enable prediction of whether a GPCR is inactive or active.

BAT output (Table 3) is similar to that from BRAT, while allowing multiple proteins to be visualized along with numeric values written by the PDB, or software such as ProFlex, in the B-value column of a PDB file.

## 4. Conclusions

This work on a set of 27 class A GPCRs presents several advances in the field of protein activity prediction that can enhance our understanding of how ligand binding affects activity:By providing a software approach not previously used to assess protein activity, ProFlex, that predicts rigid and flexible regions and their coupling within a single protein structure. This makes it unnecessary to compare protein structures, which may have a different underlying mechanism of activation. In addition, it is unnecessary to provide user-defined hypotheses regarding regions important for (in)activation. Such hypotheses can bias towards prior knowledge, and limit the understanding of regions involved in activity.Additional utilities developed here in Python, BAT and BRAT, facilitate visualizing structurally-equivalent residues in key protein regions of interest, such as binding sites or switch regions, for proteins that are sufficiently divergent that the corresponding residues cannot be defined with high confidence from sequence alignment.Although ProFlex can analyze a ligand-bound protein structure as input, in our machine learning approach, no data about the ligand or its contacts are used. Instead, ProFlex pinpoints rigid regions created by constraints within the protein’s covalent, hydrogen bond, and hydrophobic contact network, as well as separate internally rigid regions that can move relative to the protein scaffold region, followed by flexible regions.The flexibility and rigidity pattern within a protein structure defined by ProFlex can be used to create a set of features—segments of the protein labeled by their flexible, independently rigid, or mutually rigid state within the structure—that machine learning techniques such as feature selection and a classifier can use to focus down to the most discerning subset of features for predicting activity.The resulting KNN classifier of active or inactive state can drive experimental protein and ligand design, by pinpointing specific flexibility features that are more prevalent in active versus inactive structures of the protein. The KNN classifier is also intuitive, since it uses the focused feature set to identify proteins of known activity or inactivity with the most similar features to the user’s protein. This approach can also help group proteins according to similarity in the flexibility of motifs underlying (in)activation.The GPCR activity classifier using the identified six flexibility features has high accuracy: 96% correct prediction in leave-one-out cross-validation across the set of 18 inactive and 9 active GPCR structures, and 82% correct prediction when measured on held-out test sets across 10,000 iterations of bootstrap sampling. The most-predictive features colocalize around the ligand binding site proximal to the extracellular surface of the membrane protein, and thus add information to the switch regions characterized by others (such as ionic lock and tyrosine toggle), which are close to the intracellular interface with signaling partners. One of the six flexibility features, the third of helix 5 proximal to the extracellular interface, is adjacent to but non-overlapping with the transmission switch previously defined. Thus, the ProFlex-defined activation motif provides a direct connection between flexibility changes in the protein induced by ligand binding to those previously characterized in the transmission switch involving movements of helices 5 and 6 during activation.This approach can help clarify how ligand binding generates an active state in the protein. For instance, one could first use the KNN classifier step in this protocol to identify which GPCRs of known active/inactive state have the most similar flexibility state across the six key regions, relative to the user’s GPCR in complex with a designed or other test ligand. Then, the protein–protein and protein–ligand contacts in the six key regions can be compared between the user’s complex and the most similar GPCR complexes. This analysis can suggest ligand functional group changes (making or breaking specific protein contacts) to enhance the ability to inactivate (or activate) the GPCR.This intuitive feature-based classification of activity through machine learning is equally applicable to other protein families and other kinds of data. For instance, instead of ProFlex flexibility, one could test whether a subset of features defined as the presence/absence of specific residue–residue contacts (such as intraprotein hydrogen bonds, salt bridges, aromatic interactions, and/or ligand contacts) predict an active or inactive state. Because the feature selection and classifier can test many more combinations than a person could readily perform by synthesis/mutagenesis, and without bias, new information may result that usefully narrows the spectrum of experiments by homing in on key features of activation.


## Figures and Tables

**Figure 1 biomolecules-10-00454-f001:**
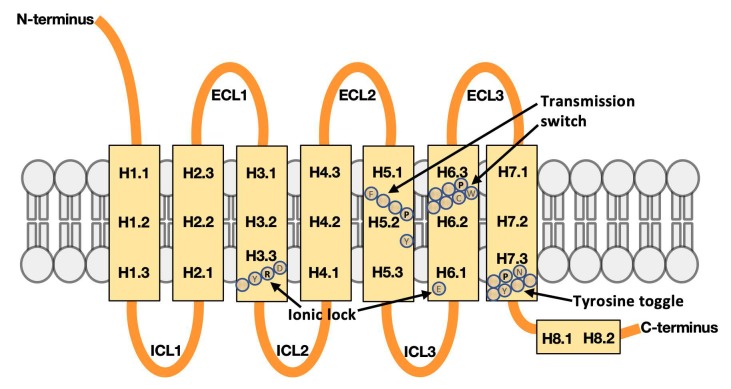
Class A GPCR architecture, partitioned into segments for machine learning analysis. Extracellular loops are labeled ECL1, ECL2, and ECL3 from N-terminus to C-terminus, and the intracellular loops are labeled ICL1, ICL2, and ICL3. Each transmembrane helix is divided into three segments, extracellular, interior, and intracellular, and indexed first by the helix number, e.g., H1, and then by the segment of helix from N-terminus to C-terminus. For instance, H1.2 is the second (interior) segment of helix 1. Helix 8, which is intracellular and shorter, was divided into two segments. Previously characterized activity switch regions and their key amino acid residues in GPCRs—the ionic lock, transmission switch, and tyrosine toggle—are also annotated [14]. The residues shown are those found in human CXCR4 (PDB entry 3ODU).

**Figure 2 biomolecules-10-00454-f002:**
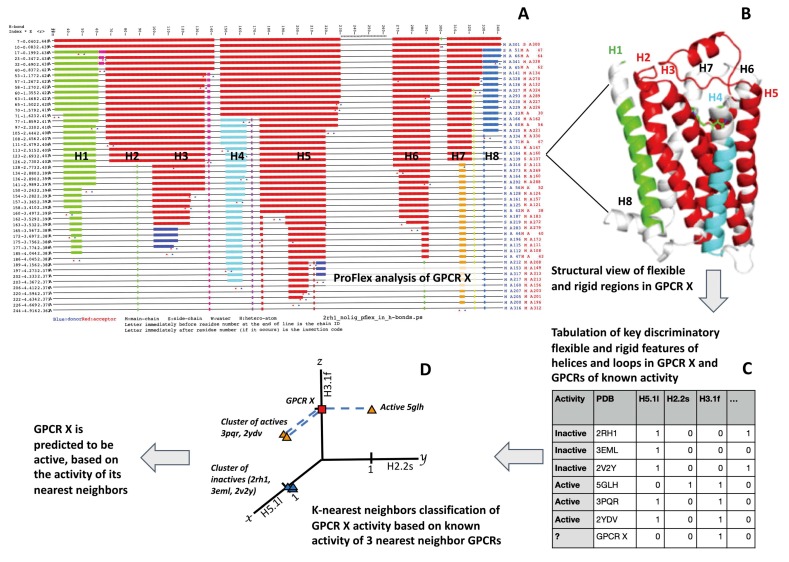
Schematic of how: (**A**) ProFlex results for a GPCR correspond to (**B**) a three-dimensional structural representation of flexibility/rigidity; (**C**) these flexibility/rigidity features are tabulated as discrete features for machine learning; and (**D**) a KNN classifier is employed with the features for activity prediction.

**Figure 3 biomolecules-10-00454-f003:**
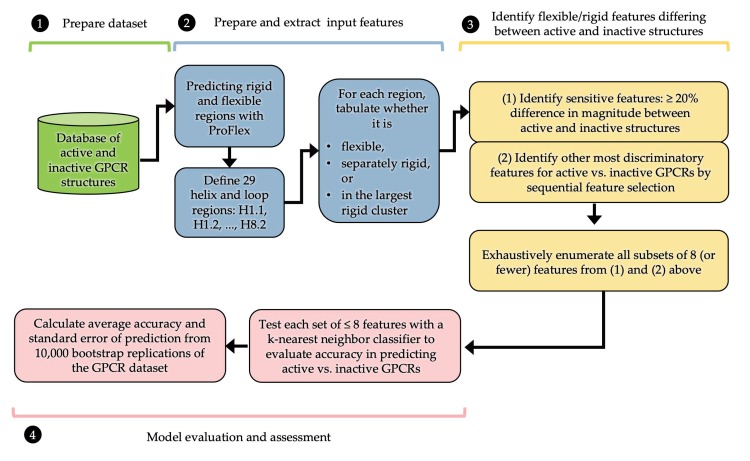
Flowchart of how ProFlex and machine learning are used to identify features that predict the active/inactive state of GPCRs from their distribution of flexible and rigid regions.

**Figure 4 biomolecules-10-00454-f004:**
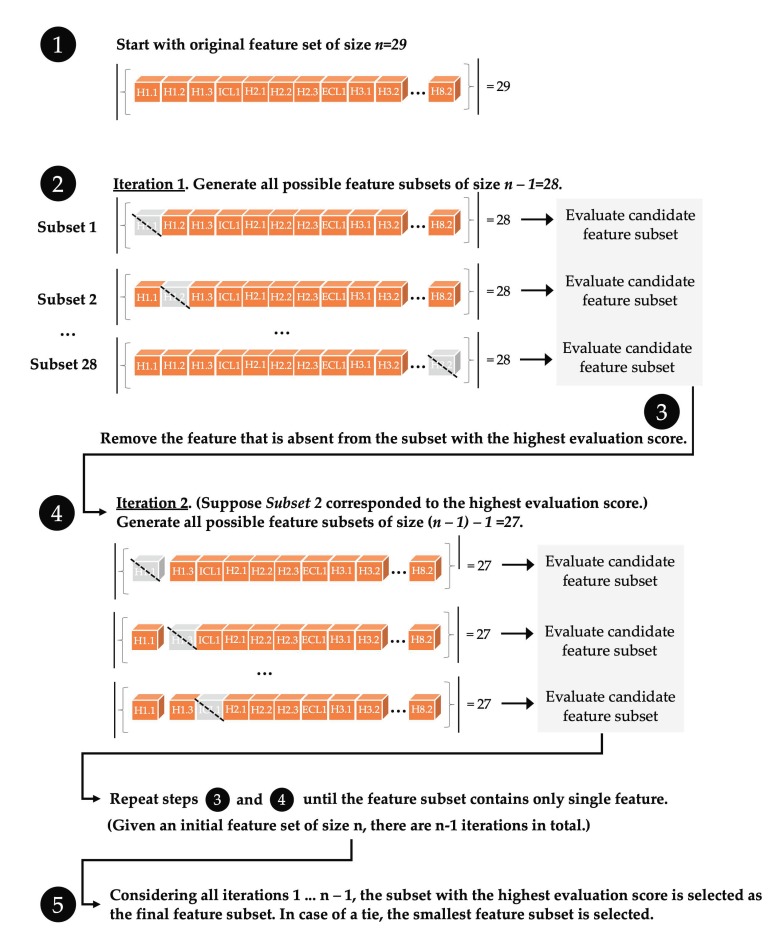
Illustration of backward sequential feature selection for identifying feature subsets that maximize the performance of a predictive model. In this study, the candidate feature subsets were evaluated by using leave-one-out cross-validation and the out-of-bag bootstrap method with a three-nearest neighbor classifier. The classifier accuracy in predicting active/inactive cases in the GPCR held-out test data was used to evaluate each feature subset, as detailed in Table 2.

**Figure 5 biomolecules-10-00454-f005:**
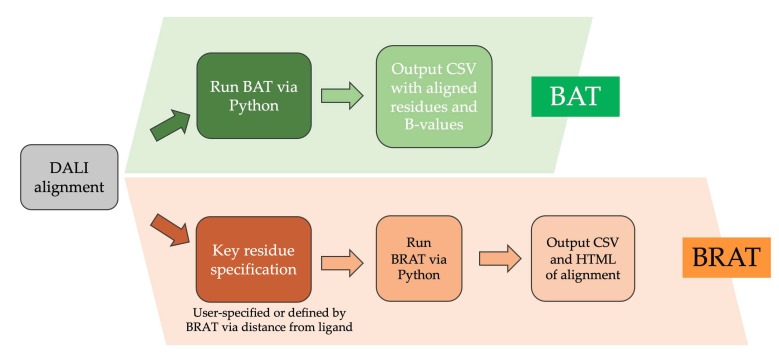
Comparison of the BRAT and BAT tools for annotating structure-based sequence alignment according to key residues (BRAT) or numeric property values from the B-value column of the PDB structure files (BAT).

**Figure 6 biomolecules-10-00454-f006:**
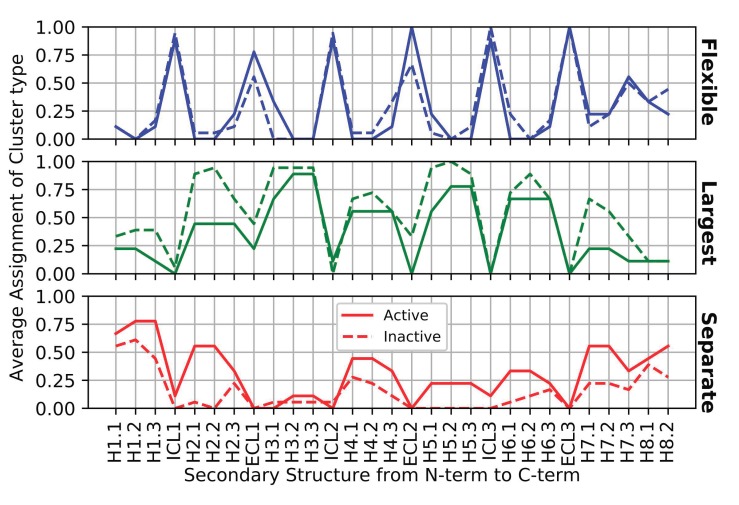
Average rigidity profiles of GPCR structures by protein segment and activity. There were 9 active structures and 18 inactive structures used for these average profiles. The occurrence values of the three rigidity assignments (f, l, and s) for active (or inactive) structures in each segment (e.g., H1.1) sum to 1.0 (100%).

**Figure 7 biomolecules-10-00454-f007:**
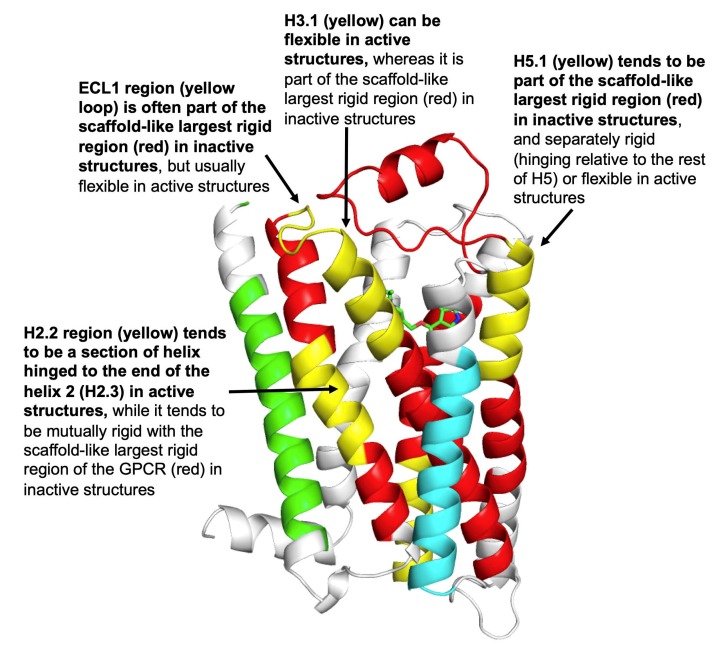
The four GPCR regions whose flexibility allows the most discrimination between active and inactive structures are highlighted in yellow; the remainder of the largest rigid region in human β2-adrenergic receptor (PDB entry 2RH1) appears in red, with two separately rigid regions in green and light blue ribbons (based on the data in Figure 2). The H2.2, ECL1, H3.1, and H5.1 segments colocalize around the ligand site, which in this case hosts the blood pressure-reducing beta-blocker, carazolol. The extracellular side of the GPCR is at the top. Trends in flexibility/rigidity of these four regions between active and inactive structures across all 27 GPCRs are annotated.

**Figure 8 biomolecules-10-00454-f008:**
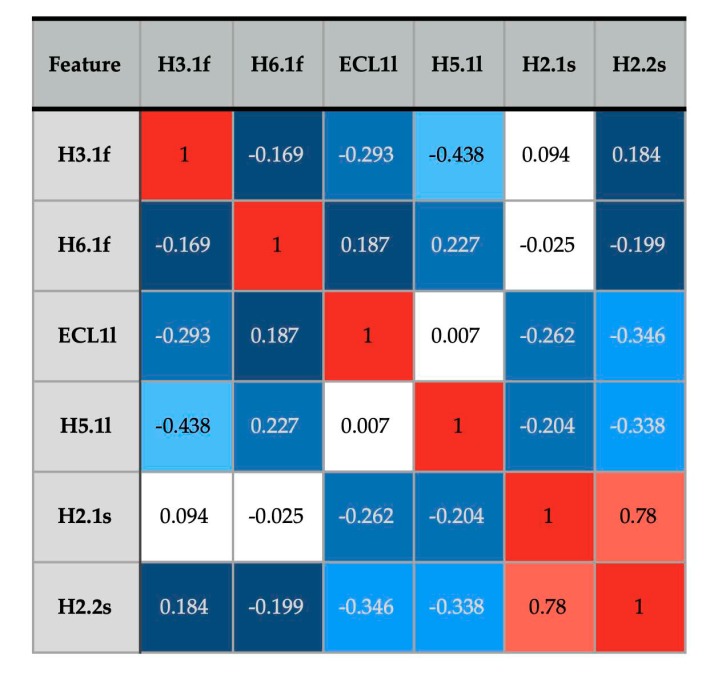
Pairwise Mathews correlation coefficient values [27] between the six features resulting in the highest-accuracy GPCR activity prediction. Absolute values near 1 reflect high correlation (e.g., feature 1 always present when feature 2 is present), values near 0 reflect a random relationship between the features, and values near -1 reflect anticorrelation (e.g., feature 1 is present when feature 2 is absent). The coloring emphasizes high correlation values in red. Decreasing correlation values are highlighted in aqua (0.3–0.5), blue (0.2–0.3), dark blue (0.1–0.2), and white (~0).

**Figure 9 biomolecules-10-00454-f009:**
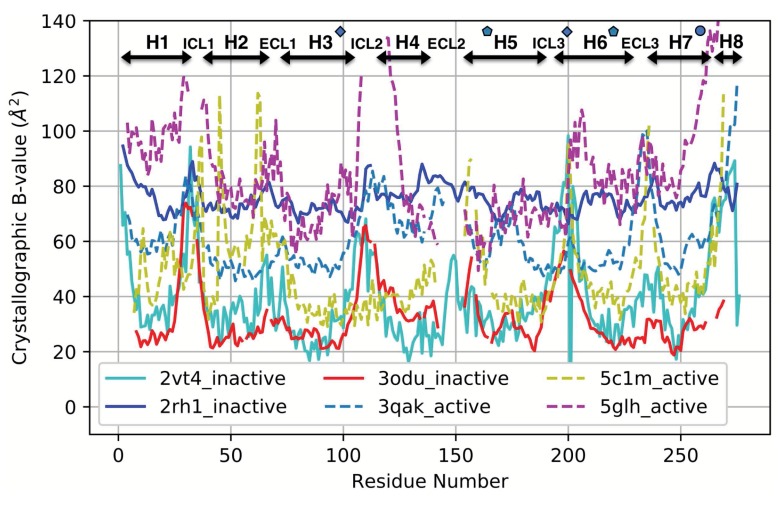
Main-chain crystallographic B-values plotted for three active (dashed line) and three inactive (solid line) class A GPCR structures. Helix, loop, and switch regions are indicated along the top of the plot, with the ionic lock residues marked by blue diamonds, the transmission switch residues marked by blue pentagons, and the tyrosine toggle region indicated by a blue circle. The structures were aligned by Dali prior to B-value comparison and indexed sequentially from the N-terminus, to avoid misalignment of structural features due to inconsistent residue numbering between GPCR structures.

**Figure 10 biomolecules-10-00454-f010:**
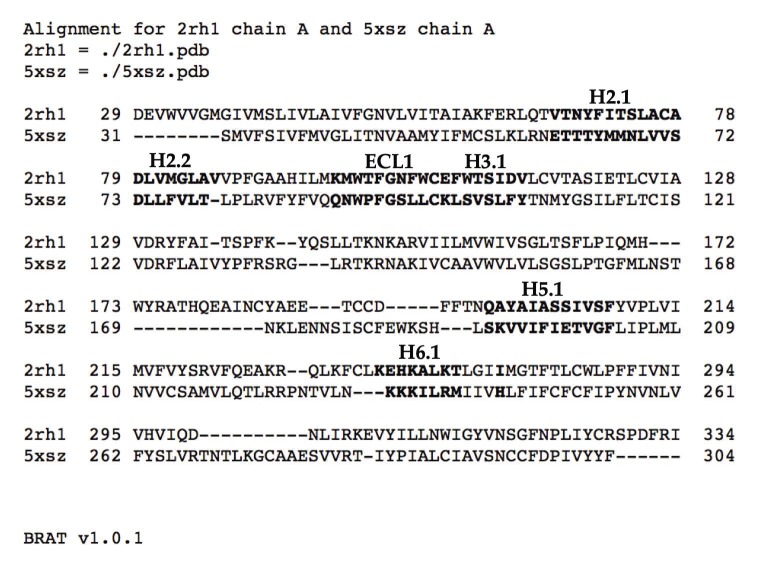
BRAT alignment (HTML view) of the sequences from PDB entries 2RH1 (human β2-adrenergic receptor) and sequence-divergent 5XSZ (zebrafish lysophosphatidic acid receptor 6; 21% identical to human β2-adrenergic receptor), which highlights in boldface the residues in 2RH1 and 5XSZ corresponding to the key features discussed in Section 3.2. Annotations of those regions appear as H2.1, etc., above the sequences.

**Table 1 biomolecules-10-00454-t001:** Crystal structures of inactive and active ligand-bound GPCRs analyzed. Ligands were removed prior to ProFlex analysis to focus on protein flexibility changes in inactive versus active proteins. See Section 2.1 for definitions of R(free) and R(work).

PDB ID	Activity *	Chain	Structure Description	Ligand Name	Organism	Resolution (Å)	R(free)	R(work)
2VT4	0	A	Beta1 adrenergic receptor	4-{[(2s)-3-(Tert-butylamino)-2- hydroxypropyl]oxy}- 3h-indole-2-carbonitrile	Meleagris gallopavo	2.7	0.27	0.21
3ODU	0	A	CXCR4 chemokine receptor	(6,6-Dimethyl-5,6-dihydro- imidazo[2,1-b][1,3]thiazol-3-yl)methyl n,n’-dicyclohexylimidothiocarbamate	Homo sapiens	2.5	0.28	0.24
3V2Y	0	A	Lyso-phospholipid sphingosine 1-phosphate receptor	{(3r)-3-Amino-4-[(3-hexylphenyl)amino]- 4-oxobutyl}phosphonic acid	Homo sapiens	2.8	0.27	0.23
3VW7	0	A	Human protease-activated receptor 1 (PAR1)	Ethyl [(1r,3ar,4ar,6r,8ar,9s,9as)-9-{(e)- 2-[5-(3-fluorophenyl)pyridin-2-yl]- ethenyl}-1-methyl-3-oxododecahydro-naphtho[2,3-c]furan-6-yl]carbamate	Homo sapiens	2.2	0.24	0.22
3EML	0	A	A2A adenosine receptor	4-{2-[(7-Amino-2-furan-2-yl[1,2,4]tria- zolo[1,5-a][1,3,5]triazin-5-yl)amino]ethyl}phenol	Homo sapiens	2.6	0.23	0.20
2RH1	0	A	Beta2-adrenergic receptor	(2s)-1-(9h-Carbazol-4-yloxy)- 3-(isopropylamino)propan-2-ol	Homo sapiens	2.4	0.23	0.20
1GZM	0	A	Bovine rhodopsin	retinal	Bos taurus	2.6	0.24	0.20
4DKL	0	A	Mu-opioid receptor	Methyl 4-{[(5beta,6alpha)-17-(cyclopropylmethyl)- 3,14-dihydroxy-4,5-epoxymorphinan- 6-yl]amino}-4-oxobutanoate	Mus musculus	2.8	0.28	0.23
3PBL	0	A	Dopamine D3 receptor	Eticlopride	Homo sapiens	2.9	0.27	0.24
4DJH	0	A	Kappa opioid receptor	JDTic	Homo sapiens	2.9	0.27	0.23
4MBS	0	A	CCR5 chemokine receptor	Maraviroc	Homo sapiens	2.7	0.26	0.22
4S0V	0	A	OX2 orexin receptor	Suvorexant	Homo sapiens	2.5	0.24	0.20
4U15	0	A	M3 muscarinic receptor	Tiotropium	Rattus norvegicus	2.8	0.26	0.23
4XNW	0	A	Purinergic receptor P2Y1	MRS2500	Homo sapiens	2.7	0.27	0.22
4YAY	0	A	Angiotensin receptor	ZD7155	Homo sapiens	2.9	0.27	0.23
4Z35	0	A	Lysophosphatidic acid receptor 1	ONO-9910539	Homo sapiens	2.9	0.27	0.28
5CXV	0	A	M1 muscarinic acetylcholine receptor	Tiotropium	Homo sapiens	2.7	0.28	0.23
5T1A	0	A	CC chemokine receptor 2 (CCR2)	BMS-681	Homo sapiens	2.8	0.27	0.23
3QAK	1	A	A2A adenosine receptor	6-(2,2-Diphenylethylamino)- 9-[(2r,3r,4s,5s)-5- (ethylcarbamoyl)-3,4-dihydroxy-oxolan-2-yl]- n-[2-[(1-pyridin-2-ylpiperidin- 4-yl)carbamoylamino]ethyl]purine- 2-carboxamide	Homo sapiens	2.7	0.27	0.22
4IAR	1	A	5-HT1b	Ergotamine	Homo sapiens	2.7	0.26	0.22
4PXZ	1	A	Purinergic receptor P2Y12 receptor	2-(Methylsulfanyl)adenosine 5’- (trihydrogen diphosphate)	Homo sapiens	2.5	0.23	0.20
2YDV	1	A	A2A receptor	n-Ethyl-5’-carboxamido adenosine	Homo sapiens	2.6	0.26	0.23
3PQR	1	A	Metarhodopsin II	retinal	Bos taurus	2.8	0.25	0.22
5C1M	1	A	Mu-opioid receptor	(2s,3s,3ar,5ar,6r,11br,11cs)-3a-Methoxy-3,14 -dimethyl-2-phenyl-2,3,3a,6,7,11c-hexahydro- 1h-6,11b-(epiminoethano)- 3,5a-methanonaphtho[2,1-g]indol-10-ol	Mus musculus	2.1	0.22	0.19
4XES	1	A	Neurotensin receptor	Neurotensin chain B	Rattus norvegicus	2.6	0.28	0.23
5GLH	1	A	Endothelin receptor type B	Endothelin-1 peptide chain B	Homo sapiens	2.8	0.28	0.23
5TVN	1	A	5-HT2b receptor	LSD	Homo sapiens	2.9	0.26	0.21

* 1 = active; 0 = inactive.

**Table 2 biomolecules-10-00454-t002:** Accuracy of the highest-performing feature sets upon KNN classifier assignment of active or inactive state from leave-one-out and bootstrap testing on subsets of 27 GPCR structures.

Feature Set	Leave One Out Accuracy	Bootstrap Mean Accuracy	Standard Error
ECL1l, H2.2s, H3.1f, H5.1l	96.3%	79.6%	14.0%
ECL1l, H2.1s, H2.2s, H3.1f, H5.1l, H6.1f	96.3%	81.7%	12.6%
ECL1l, H2.2l, H2.2s, H3.1f, H5.1l, H6.1f	96.3%	81.1%	12.5%
ECL1, H2.1l, H2.2s, H3.1f, H5.1l, H6.1f	96.3%	80.8%	12.5%
Dummy classifier: always predicts inactive	66.6%	60.4%	15.5%

**Table 3 biomolecules-10-00454-t003:** BAT output showing the alignment of five GPCR structures from the PDB (2RH1, 3ODU, 3QAK, 5C1M, and 5GLH) from their Dali structural alignment with 2VT4 (β1-adrenergic receptor). Values from the B-value column, in this case containing the flexibility index value written by ProFlex (where 0 is most rigid and larger is more flexible), are also aligned.

PDB Entry	Structurally Aligned Residues										
	Chain ID, Residue Number	C39	C40	D41	D42	D43	D44	D45	D46	D47	…
**2vt4**	Residue Name	GLN	TRP	GLU	ALA	GLY	MET	SER	LEU	LEU	…
	Flexibility Index	87.52	65.84	70.48	55.27	56.21	45.47	40.01	35.27	41.46	…
	Chain ID, Residue Number	-	A32	A33	A34	A35	A36	A37	A38	A39	…
**2rh1**	Residue Name	-	TRP	VAL	VAL	GLY	MET	GLY	ILE	VAL	…
	Flexibility Index	-	94.6	91.1	87.62	86.08	84.86	82.58	80.88	79.67	…
	Chain ID, Residue Number	-	-	-	-	-	-	-	B43	B44	…
**3odu**	Residue Name	-	-	-	-	-	-	-	THR	ILE	…
	Flexibility Index	-	-	-	-	-	-	-	27.28	25.37	…
	Chain ID, Residue Number	-	-	-	A7	A8	A9	A10	A11	A12	…
**3qak**	Residue Name	-	-	-	SER	VAL	TYR	ILE	THR	VAL	…
	Flexibility Index	-	-	-	69.69	67.31	61.19	59.49	60.78	58.29	…
	Chain ID, Residue Number	-	-	-	-	-	-	B72	B73	B74	…
**5c1m**	Residue Name	-	-	-	-	-	-	ARG	ASP	VAL	…
	Flexibility Index	-	-	-	-	-	-	34.11	42.58	40.51	…
	Chain ID, Residue Number	-	-	-	A102	A103	A104	A105	A106	A107	…
**5glh**	Residue Name	-	-	-	TYR	ILE	ASN	THR	VAL	VAL	…
	Flexibility Index	-	-	-	3.03	92.12	99.68	98.95	1.22	94.85	…

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
