# Peer review of "Machine Learning to Identify Flexibility Signatures of Class A GPCR Inhibition"

_biomolecules, 2020, doi:10.3390/biom10030454_

Round 1

Reviewer 1 Report

In this manuscript, Bemister-Buffington et al., have developed a classification model to predict and distinguish active vs inactive forms for Group A GPCRs employing a machine learning method - k-neighbour classification based on ProFlex analysis of rigid and flexible regions. They then develop BRAT and BAT programs to visualize the segments responsible for active and inactive states as well as B-factor values in the sequence alignment. The manuscript is of a good standard and meets the scope of the journal. I recommend publishing after addressing the points below:

  1. The author should link findings of correlated domains with experimental information around the mechanism of activation and mutations for the investigated receptors. Good to have a couple of receptor cases and point out the application of classification in highlighting areas and important residues in the receptors. At the moment it is very general and not clear how it will be useful in hypothesis generation for experimentalists willing to find key residues for stabilizing one or another receptor state.
  2. Along the same lines the authors might want to consider to explore case studies for one and two receptors with unavailable structural information but available indirect studies (mutations) by constructing homology models and evaluating the performance of the model.
  3. The authors could elaborate on why only the KNN classifier was used and cross-test with other methods were not investigated. 
  4. The authors should explain R(free) and R(work) abbreviation in Table 1 captions.
  5. In Figure 2A the resolution should be improved.

Author Response

1) The author should link findings of correlated domains with experimental information around the mechanism of activation and mutations for the investigated receptors. Good to have a couple of receptor cases and point out the application of classification in highlighting areas and important residues in the receptors. At the moment it is very general and not clear how it will be useful in hypothesis generation for experimentalists willing to find key residues for stabilizing one or another receptor state.

Response: There is a vast GPCR activation literature from the recent accomplishments of cryo-electron microscopy researchers and X-ray crystallography and molecular dynamics labs, which report individual observations on conformational and bond transitions that occur between inactive and active states of GPCRs.  To summarize the key features of the data for some of the 27 GPCR structures studied here and compare them with ProFlex machine learning observations (especially given the different types of data collected by different labs), would be a significant endeavor.  Given the current length of the manuscript and its focus on a new way of predicting active versus inactive states, we fear that including such a comparison could not be completed within the short timeframe for revision and would also bury the main points of the paper.  However, we see it would be helpful to briefly compare ProFlex results with experimental results for a well-studied protein, and point out what the current analysis can add to already published work.  Therefore, we have added the following paragraph to line 544 (section 3.3):

“How can these concepts and methods be applied to an individual GPCR, to help define residues that contribute to activation or inactivation?  Here we focus on one of the best-studied GPCRs, rhodopsin, given structures of its inactive (PDB entry 1GZM) and active (PDB entry 3PQR) states.  Of the six key flexibility features identified here across class A GPCRs, two differ significantly between inactive and active rhodopsin states. ProFlex results show that the H2.1 region (the cytoplasmic third of helix 2) is part of the largest rigid region in inactive rhodopsin (known as opsin), while its initial residues are flexible in the active form.   Secondly, the H6.1 region (the cytoplasmic third of helix 6) is separately rigid in opsin, while being flexible in active rhodopsin.  These flexibility changes are consistent with the trend of key regions in class A GPCRs, as a whole, to contribute to the largest rigid region (the protein scaffold) in the inactive state while exhibiting increased flexibility and uncoupling to other regions of the protein in the active state (Figure 7).  How do the ProFlex results compare with experiments characterizing the rhodopsin transition between inactive and active states?  H6.1 includes the ionic lock residue E247, which forms a salt bridge with R135 in opsin but not in active rhodopsin. The loss of the ionic lock interaction is consistent with the ProFlex observation of increased flexibility in H6.1 upon activation.  Secondly, the cytoplasmic end of helix 6 (H6.1) is observed to hinge towards helix 5 upon comparison of the active and inactive structures, which is consistent with the increased flexibility of H6.1 found by ProFlex in the active state. Narrowing down the most important protein flexibility transition sites for activation from the plethora of conformational changes observed between crystal structures is a valuable application for ProFlex machine learning.  This can suggest a much more focused set of experiments - to test H6.1 hinge residues, for instance - as well as indicating which flexibility transitions are shared with other class A GPCRs.”

2) Along the same lines the authors might want to consider to explore case studies for one and two receptors with unavailable structural information but available indirect studies (mutations) by constructing homology models and evaluating the performance of the model.

Response: Given the short timeline for revisions to the paper, applying the ProFlex machine learning approach to homology models lies beyond the scope of the current paper.  Homology models for GPCRs contribute importantly to the field, given the difficulty of preparing native-like, pure membrane proteins for experimental structural determination.  However, in past work, we noted that homology modeling does not always provide precise enough locations for the donor and acceptor atoms of hydrogen bonds, resulting in fewer identified bonds and underconstrained, overly flexible results from ProFlex.  However, other aspects of a protein structure that are less dependent on positional resolution, such as the spatial location of different amino acid types and their clustering in protein structures, could also be good predictors of sites important for protein activation.  Such alternative types of data as features can be used and tested as predictors with the same machine learning approach.  The above paragraph has been added to section 2.2 at line 201.

3)  The authors could elaborate on why only the KNN classifier was used and cross-test with other methods were not investigated.

Response: This is a good point that we now address. The primary reasons for choosing a KNN classifier are its robustness and interpretability. In particular, we wanted to avoid machine learning algorithms that require extensive parameter tuning, which can more readily result in overfitting when working with small datasets. Additionally, we wanted to use a model that is intuitive, interpretable, and can be updated when more structures become available in the future.  Alternatives to KNN were tested, as well, and have added information about them to the end of section 2.4 (line 412), as follows:

“The general exhaustive and sequential feature selection methods outlined in this section can be combined with any machine learning algorithm for classification, and the specific MLxtend software implementation of SFS and EFS used in this study is compatible with any classifier implemented in Scikit-learn. We repeated the steps outlined in this section using generalized linear models such as logistic regression and a linear support vector machine (SVM) instead of KNN.  Both logistic regression and linear SVM resulted in feature subsets with lower predictive performance compared with the KNN classifier, which is likely due to the linear models' inability to capture the complex relationship between the input features and the class labels. A nonlinear radial basis function (RBF) kernel SVM was not considered in this study, as it requires extensive hyperparameter tuning and is thus prone to overfitting on a small dataset such as ours.  Finally, we chose and focused on KNN as the primary classifier for this study, because it does not require extensive hyperparameter tuning and remains interpretable; for instance, predictions for new structures can be analyzed by querying and analyzing its nearest-neighbor structures in the existing data set.”

4) The authors should explain R(free) and R(work) abbreviation in Table 1 captions.

Response:  The following information has been added to the text (line 131, section 2.1) and is now referenced in the table caption (line 150): “Crystallographic R values measure the percentage difference in electron density when the data gathered from the diffraction experiment is overlaid with the electron density calculated from the atomic model that was fit into the electron density by the crystallographer, based on the known number of electrons associated with each atom type.  A problem with this R(work) definition for assessing structural quality is that the refinement software used in structure determination is often designed to improve the fit between the model and the experimental electron density, which improves (lowers) the R(work) value but introduces bias. The R(free) value is used as a less biased measure of structural agreement with the electron density data.  To calculate R(free), 10% of the experimental observations are removed from the data set before refinement, and the refinement is then carried out with the remaining 90%. The R(free) value, also reflecting the percent difference in electron density between the experimental data and fitted model, is measured by comparing the electron density of the model fitted and refined to the 90% data set with the experimental electron density calculated from the held-out 10% of the data.   For an ideal structure, the R(free) value is close to the R(work) value, though typically it is higher.  Lower values for both R(free) and R(work) are more favorable, showing higher agreement between the experimental data and the atomic structure (https://pdb101.rcsb.org/learn/guide-to-understanding-pdb-data/r-value-and-r-free).”

5)  In Figure 2A the resolution should be improved.

Response: We have replaced Figure 2A at line 259 by a high-resolution vector graphics version that allows magnification without loss in quality.

Reviewer 2 Report

The presented method is reasonable and the leave-one-out and bootstrap-sampling validation seems appropriate. It is highly expected that it works fine for structures similar to the current set of 27 GPCR structures. However, it is not guaranteed that it can be unconditionally extrapolated into other GPCRs. Some statements on limitation and/or to applicability to GPCRs other than the current data set are wanted. Applicability to other protein families (Lines 635-642) is also an important topic, and working conditions for the proposed method, if any, would be interesting for readers.

Concerning the BRAT tool, it would be also useful if some (rough) prediction for proteins without structural data is possible; if any, please state possible directions.

Minor points:

Line 137: "to" is repeated.

Line 591: "comma separated value" here seems redundant.

Author Response

1) Some statements on limitation and/or to applicability to GPCRs other than the current data set are wanted.  Applicability to other protein families (Lines 635-642) is also an important topic, and working conditions for the proposed method, if any, would be interesting for readers.

Response:  Aside from the structural resolution caveat, there is no fundamental limitation to the application of this method to any protein family for which 3D structures and at least one known active case and one inactive case are available.  That said, we would not advise mixing GPCRs from different families together, because the structures between GPCR families differ, as do their molecular partners and mechanisms of activation (especially for GPCRs that bind ligands in an extracellular domain).  In different GPCR families, a different set of features may be key to activation.  They can be unveiled by the machine learning feature selection approach described here when trained on that particular family.  Another aspect that can vary from family to family is whether the automatically chosen ProFlex energy level (HETHER option C, mentioned above in section 2.3) is appropriate for that particular family.  This can be assessed most readily by a user knowledgeable about the protein family, by inputting to ProFlex a well-characterized active structure, then a well-characterized inactive structure, and visually identifying the energy level in the two ProFlex hydrogen bond dilution profiles (e.g., Figure 2A) that best identifies the known (literature-described) flexibility features that differ between the active and inactive states. Once that energy level is established, ideally by evaluating more than one protein in the family, it can be used as the energy threshold for predicting the active/inactive state of other family members. Because known exemplars of active and inactive states are used by the KNN classifier as the basis for predicting the activity of new structures, including more known examples may also improve the predictive accuracy. The preceding paragraph has been added to the Methods section at line 210.

2) Concerning the BRAT tool, it would be also useful if some (rough) prediction for proteins without structural data is possible; if any, please state possible directions.

Response: A meaningful comparison between key regions in two proteins (e.g., ligand binding or allosteric pathway residues) depends upon a reliable alignment of their protein sequences, rather than requiring 3D structures.  For the present work, we focused on structure-based alignments because they allow definition of a clear correspondence between residues in protein regions where the sequence similarity is too low to allow confident sequence alignment.   The helpfulness of structure-based alignment is particularly clear for the ligand binding sites of different class A GPCRs, which bind remarkably diverse ligands and therefore are not well conserved in sequence, while being substantially conserved in 3D structure.  Structural alignment can define which residues between two proteins occur in the same position in the structure (or not).  Sequence alignment methods that align one sequence to a multiple sequence alignment for the protein family, where the constituent sequences are chosen to reflect the protein’s evolutionary diversity, can partially address the challenge of aligning divergent sequences.  This is because multiple sequence alignments containing many evolutionarily related sequences implicitly include information about the tolerance for different amino acid mutations and insertions or deletions at each position, which allows the alignment method to knowledgeably penalize for the presence of improbable residues or insertions or deletions at each position.  For low-identity regions, it is still important to evaluate a local measure of the likelihood that each region of the sequence is correctly aligned before considering the residues in the proteins to be equivalent.  Once such an alignment is available from any robust approach, formatting it like the standard Dali input (see documentation under https://github.com/psa-lab/ Protein-Alignment-Tool) will allow BRAT and BAT to be run successfully.  This paragraph has been added to the end of section 2.5, at line 462.

3) Line 137: "to" is repeated.

Response:  Thank you for noting this; it has been corrected (now at line 156).

4) Line 591: "comma separated value" here seems redundant. 

Response:  We have removed this redundancy in the first phrase of the Figure 11 caption (now at line 711) for smoother reading.